# A Comparative Study of the Effects of Whole Cereals and Refined Cereals on Intestinal Microbiota

**DOI:** 10.3390/foods12152847

**Published:** 2023-07-27

**Authors:** Dan Yu, Li Zhu, Minjie Gao, Zhongwei Yin, Zijian Zhang, Ling Zhu, Xiaobei Zhan

**Affiliations:** 1Key Laboratory of Carbohydrate Chemistry and Biotechnology, Ministry of Education, School of Biotechnology, Jiangnan University, Wuxi 214122, China; 6210207009@stu.jiangnan.edu.cn (D.Y.); zhanzhuli@yahoo.com (L.Z.); jmgao@jiangnan.edu.cn (M.G.); 7200201071@stu.jiangnan.edu.cn (Z.Y.); zzjyyyyyg@163.com (Z.Z.); zhuling_mimg@outlook.com (L.Z.); 2A & F Biotech. Ltd., Burnaby, BC V5A 3P6, Canada

**Keywords:** refined flour, whole wheat flour, intestinal microbiota, short-chain fatty acid

## Abstract

Cereals are one of the most important foods on which human beings rely to sustain basic life activities and are closely related to human health. This study investigated the effects of different steamed buns on intestinal microbiota. Three steamed buns were prepared using refined flour (RF), 1:1 mixed flour (MF), and whole wheat flour (WF). In vitro digestion simulations were conducted using a bionic gastrointestinal reactor (BGR) to examine their influence on intestinal microbiota. The results showed that at 0.5% addition, butyric acid and short-chain fatty acids in WF were significantly different from those in RF and MF (*p* < 0.05). WF also promoted the proliferation of beneficial microbiota, such as *Megamonas* and *Subdoligranulum*. At 0.5%, 1.0%, and 1.5% additions of WF, acetic acid and short-chain fatty acids at 1.5% WF increased by 1167.5% and 11.4% from 0.5% WF, respectively, and by 20.2% and 7.6% from 1.0% WF, respectively. WF also promoted the proliferation of *Bifidobacterium*, *Lactobacillus*, and *Bacteroides* and inhibited the growth of pathogenic microbiota, such as *Streptococcus*, *Enterococcus*, and *Klebsiella*. These findings support the consumption of whole cereals and offer insights into the development of new functional foods derived from wheat.

## 1. Introduction

Cereals contribute approximately 50.0% of global calorie intake. In low- and middle-income countries such as those in Africa and South Asia, this proportion is even as high as 70.0% [1]. Studies have shown a positive association between high consumption of refined cereals with a high glycemic index and the incidence of cardiovascular disease, and a high intake of refined cereals is associated with a high risk of total mortality and cardiovascular disease. A significant linear association exists between high consumption of refined cereals and total mortality, with each 200 kcal increase in refined cereals intake associated with an approximate 3.0% increase in the risk of death [2]. Several prospective cohort studies have reported on the effects of whole versus refined cereals on human mortality and chronic disease. A study showed that for every three additional servings of whole cereals per day, mortality and cardiovascular mortality were reduced by 19.0% and 26.0%, respectively [3]. A meta-analysis showed that the intake of refined cereals ranged from 15 g/day to 540 g/day; when comparing the high- and low-intake groups, the high-intake group had a significantly increased risk of coronary heart disease [4]. The association between refined cereals intake and clinical outcomes varies across regions. It is particularly pronounced in China, possibly due to the high and variable intake of refined cereals in the Chinese population [5]. With socioeconomic development and lifestyle changes, a major shift has been found in the dietary structure of the Chinese population, especially with the increased intake of wheat. Wheat consumption in China has steadily increased over the past 20 years and is positively associated with cardiovascular disease risk [6]. Thus, studies on the health effects of whole cereals and refined cereals are crucial.

Regular consumption of whole cereals, vegetables, and fruits rich in dietary fiber can help prevent and treat chronic metabolic diseases [7,8]. Many studies have shown that the development of chronic metabolic diseases is closely linked to changes in intestinal microbiota. Short-chain fatty acids (SCFAs) are one of the metabolites of the intestinal microbiota and can affect host metabolism through a variety of pathways, including energy metabolism, insulin metabolism, and cholesterol metabolism [9]. Dietary fiber and intestinal-microbe-derived SCFAs exert a variety of beneficial effects on host energy metabolism not only by improving the intestinal environment but also by directly affecting various peripheral tissues of the host [10,11]. Whole cereals are rich in dietary fiber, resistant starch, and oligosaccharides that can be fermented and used by intestinal microbiota, thereby affecting consumer health [12]. There is growing evidence from human and animal studies suggesting that whole cereal consumption can affect gut microbiota composition and stimulate gut-derived SCFAs and hormones involved in appetite regulation [13]. Intake of whole cereals has consistently been associated with a reduced risk of cardiovascular disease, obesity, type 2 diabetes, and certain types of cancer [14]. In a study involving 50 obese participants, the intake of whole wheat bread resulted in a significant and safe reduction in visceral fat area when compared to refined wheat bread (*p* < 0.05) [15]. A double-blind randomized crossover trial indicated that participants who consumed whole wheat pasta for 7 days experienced appetite control in comparison to the refined wheat pasta group [16]. In another study, refined rice, refined wheat, unrefined rice, and whole wheat were fed to normal rats, and the results showed that whole wheat modulated the composition of intestinal microbiota and increased the concentration of SCFAs in rats [17].

Whole cereals are defined by the American Association of Cereal Chemists International and the FDA as consisting of intact, ground, cracked, or flaked fruit of the grain the principal components of which—the starchy endosperm, germ, and bran—are present in the same relative proportions as they exist in the intact grain [18]. Wheat is mainly composed of endosperm, germ, and bran, which contain protein, minerals, vitamins, and digestible carbohydrates in addition to a large amount of dietary fiber. The non-digestible carbohydrates from dietary fiber are the main source of energy for intestinal microbiota [9,10,11,19]. By processing wheat in different ways, refined flour (RF) and whole wheat flour (WF) can be obtained. RF is the flour obtained by refining after removing the bran and germ, so it lacks many nutrients [20,21]. WF is a flour made from whole wheat that contains both bran and germ, so it is rich in dietary fiber, lipids, minerals, and bioactive compounds that are beneficial to human health.

The prospective cohort studies described above can accurately investigate the effects of food on human health, but they are time-consuming and expensive. The bionic gastrointestinal reactor (BGR) is a quasi-realistic simulation of the human digestive system; the BGR facilitates in vitro digestive simulations, and its advantages include simplicity, safety, and reproducibility [22,23]. Minimal research has been reported on the changes in the composition of steam buns made from different types of flour during the in vitro simulation of gastrointestinal digestion. In this paper, RF and WF were processed into different types of steamed buns, namely, RF, 1:1 mixed flour (MF), and WF. Their compositional changes in an in vitro digestion model were analyzed by in vitro simulated digestion experiments using the DNS method for reducing sugars, anthracene–sulfate method for total sugars, and water-soluble ABTS method. The changes in pH, OD_600_, SCFAs, and intestinal microbiota were examined by using in vitro fecal fermentation to investigate the effects of whole cereals and refined cereals on intestinal microbiota by using a short and efficient method for cereal evaluation.

## 2. Materials and Methods

### 2.1. Materials

WF and RF were purchased at a local supermarket. Simulated gastric fluid and simulated small intestine fluid were purchased from Nanjing Xinfan Biotechnology Corporation, Nanjing, China. ABTS was purchased from Beijing Yinuokai Technology Corporation, Beijing, China. Microporous filter membrane (0.22 μm diameter) was purchased from Nantong Sea Star Equipment Corporation, Nantong, China.

### 2.2. Fecal Sources and Strain Preservation

Fecal preservation was performed according to Aguirre et al. [24]. Fresh fecal samples were obtained from four healthy volunteers who followed a normal Chinese diet, were free of digestive disease, and had not received antibiotics for at least three months. The fresh fecal samples collected were weighed individually to the same weight, mixed 1:1 with sterile dialysis solution, stirred to homogenize, and added with glycerol as a cryoprotectant to yield a final concentration of 12.0%–15.0% (*w*/*w*). The samples were filtered through four layers of sterile gauze, divided into individual aliquots of the same volume, flash frozen in liquid nitrogen, and stored at −80 °C until use [19,25].

### 2.3. Steamed Bun Preparation

In this experiment, three types of steamed buns were made using different ratios of flour: WF:RF at 0:1, 1:1, and 1:0. The ingredients used for each type of steamed buns were 50.0 g of flour, 0.5 g of yeast, and 22.0 g of water. The dough was rolled out into long strips, folded, stretched three to five times to make the surface smooth, rolled up, and cut into small pieces as steamed buns. The steamed buns were fermented for 30–50 min until light and fluffy and then steamed in boiling water for 20–25 min.

### 2.4. In Vitro Simulation of Oral, Gastric and Small Intestinal Digestion

The steamed buns were subjected to simulated saliva digestion with reference to the method of Mulet-Cabero et al. [26]. The simulated saliva was composed of 6.2 g/L NaCl, 2.2 g/L KCl, 1.5 g/L CaCl_2_·2H_2_O, and 5.5 mg/L α-amylase. As buns are solid foods that need to be chewed, we used a commercially available electric blender to chop the buns first and then mixed 1:1 (*w*/*w*) with simulated saliva and placed in a shaking water bath (80 r/min) at 37 °C for 40 s. The mixture was then heated at 100 °C for 10 min to inactivate α-amylase and terminate the reaction.

A new in vitro digestion model device developed in the laboratory, a BGR, was used for this experiment. The device simulated intestinal peristalsis and temperature (37 °C) by controlling the flow of warm water between a glass jacket and a flexible hose through a water circulation device. A 10.0% volume of simulated gastric fluid was first pumped into the BGR, and the sample was pumped in at a certain flow rate. The simulated gastric fluid secretion flow rate was simultaneously adjusted from 0.6 mL/min to 3.2 mL/min from 0 min to 40 min and back down to 0.6 mL/min from 40 min to 60 min, resulting in a final sample to simulated gastric fluid ratio of 1:1 (*v*/*v*). The pH was also controlled by a pH sensor and HCl solution (0.5 mol/L) at around 2.0. The peristaltic frequency of the stomach was adjusted to 3 times/min, and digestion was carried out at pH 2.0 for 2 h. Sampling was carried out every 30 min for the determination of reducing sugars, total sugar content, and ABTS free radical scavenging capacity. Finally, the remaining gastric digest was heated at 100 °C for 10 min to inactivate the enzyme and terminate the reaction.

A peristaltic pump at the end of the gastric reactor was connected to a bionic small intestine reactor containing a certain volume of simulated small intestine fluid. The gastric contents were gradually delivered to the small intestine reactor and simultaneously pumped into the simulated small intestine fluid so that the two were mixed at a ratio of 1:3 (*v*/*v*). The pH was simultaneously controlled at around 7.0 by using a pH sensor and NaOH solution (0.5 mol/L). The peristaltic frequency of the small intestine was adjusted to three times/min, and digestion was carried out at pH 7.0 for 4 h. Sampling was performed every 1 h for the determination of reducing sugars, total sugar content, and ABTS free radical scavenging capacity. The remaining small intestine digest was heated at 100 °C for 10 min to inactivate the enzymes, freeze-dried into a powder, and stored at 4 °C to be used as a carbon source for in vitro fecal fermentation on intestinal microbiota [27].

### 2.5. In Vitro Fecal Fermentation

The experiments referred to and modified the in vitro fecal fermentation method of Yang et al. [28], where fermentation was carried out in an anaerobic incubator. Fermentation was carried out using basal medium (g/L): carbon source, tryptone (2.0), yeast extract (2.0), NaCl (0.1), K_2_HPO_4_ (0.04), KH_2_PO_4_ (0.04), MgSO_4_·7H_2_O (0.01), CaCl_2_·6H_2_O (0.01), NaHCO_3_ (2.0), heme chloride (0.025), vitamin K_1_ (0.002; filtered to remove microbiota), L-cysteine hydrochloride (0.5), bile salts (0.5), edged aspartame solution (0.25), and 0.01 (*v*/*v*) Tween 80. The medium had a pH of 6.8–7.0, and it was sterilized at 115 °C for 30 min. Two groups were designed: (1) freeze-dried samples of RF, MF, and WF digests were used as carbon sources (added at 0.5%, *m*/*v*) and fermented separately to screen the optimal carbon source; (2) the optimal carbon source was used as a variable and added at 0.5%, 1.0%, and 1.5% (*m*/*v*) to explore the effect of different additions on the intestinal microbiota.

The anaerobic tubes were filled with 5.4 mL of basal medium each in a working volume of 6.0 mL. The frozen microbiota solution was thawed in a 37 °C water bath for 30 min. Subsequently, 0.6 mL of microbiota solution was inoculated into the medium and incubated anaerobically at 37 °C for 48 h. All the above aseptic operations were carried out in an anaerobic incubator, and each fermentation system was performed in triplicate. Samples were obtained every 12 h during fermentation, snap-frozen in liquid nitrogen, and stored at −80 °C in a refrigerator until analysis. Samples that were snap-frozen in liquid nitrogen immediately after inoculation were used as controls (0 h of fermentation).

### 2.6. DNS Method for Reducing Sugar Content

The content of reducing sugars was determined using the DNS method. The DNS reagent was first prepared: 6.3 g of 3,5-dinitrosalicylic acid and 262.0 mL of 2.0 mol/L NaOH were added to a 500.0 mL solution containing 182.0 g of potassium sodium tartrate. Approximately 5.0 g of heavy phenol and 5.0 g of Na_2_SO_3_ were added to the solution, which was stirred to dissolve, cooled, fixed with water to 1000.0 mL, and stored in a brown bottle for 1 week before use. The glucose standard curve was then prepared: 0.0, 0.2, 0.4, 0.6, 0.8, and 1.0 mL of glucose standard solution (1.0 mg/mL) was placed into test tubes and made up to 1.0 mL with deionized water. About 1.5 mL of DNS reagent was added to each test tube. The tubes were boiled for 5 min, self-cooled at room temperature, and made up to 10.0 mL with water. The absorbance was measured at 540 nm, and the standard curve was plotted. Final determination of reducing sugars: 1.0 mL of suitably diluted sample was placed in a test tube, and the above operation was repeated. The absorbance was measured at 540 nm, and the reducing sugar content was calculated from the standard curve.

### 2.7. Anthracene–Sulfate Method for the Determination of Total Sugars

The total sugars content was determined using the anthracene–sulfate method [29]. First, an anthracene–sulfate solution was prepared: 0.1 g of anthracene was dissolved in 100.0 mL of 76.0% sulfuric acid solution. The glucose standard curve was then prepared by transferring 0.0, 0.2, 0.4, 0.6, 0.8, and 1.0 mL of glucose standard solution (1.0 mg/mL) into test tubes and adding distilled water until reaching a volume of 1.0 mL. About 5.0 mL of anthracene–sulfate solution was added accurately to each tube, boiled for 10 min, and cooled at room temperature for 10 min. The absorbance was measured at 620 nm, and the standard curve was plotted. For the final determination of total sugars, 1.0 mL of the sample was diluted in a test tube and treated in the same way.

### 2.8. Determination of ABTS Free Radical Scavenging Capacity

The ABTS free radical scavenging capacity was measured according to the method of Denaro et al. [30]. Potassium persulfate (4.3 mmol/L) and ABTS (1.7 mmol/L) were mixed at a ratio of 1:5 (*v*/*v*) and stored in a brown reagent bottle for 12–16 h. The ABTS solution was diluted in deionized water to give an absorbance value of 0.7 ± 0.02 at 734 nm before use. About 4.0 mL of ABTS radical solution was added to 200.0 μL of the simulated digestion solution, and the absorbance was measured at 734 nm after the reaction for 6 min at room temperature and protected from light.

The clearance (%) of the samples was calculated as follows:Clearance(%)=(A0−A)/A0×100%

In the formula, A is the absorbance of the sample with ABTS radical solution, and A0 is the absorbance of ABTS radical solution with deionized water.

### 2.9. Measurement of SCFAs

The yields of SCFAs were analyzed by using gas chromatography (GC) according to the experimental method of Wu et al. [31]. The contents of acetic acid, propionic acid, isobutyric acid, butyric acid, isovaleric acid, and valeric acid were also determined. First, 1.0 mL of fermentation broth was removed from each sample to be tested after centrifugation at 12,000 r/min for 5 min. About 10.0 μL of internal standard (100.0 mmol/L 2-ethylbutyric acid), 250.0 μL of HCl, and 1.0 mL of anhydrous ethyl ether were added to extract the target product. The organic phase was separated by vortexing for 3–5 min, de-watered with anhydrous sodium sulfate, and filtered through a 0.22 μm pore size organic membrane.

The SCFAs were then analyzed by using GC (Agilent Technologies, Inc. -7890A, Santa Clara, CA, USA) equipped with an HP-INNOWAX column. The oven temperature was 60 °C, which was ramped up to 190 °C within 4 min. The injector temperature was set to 220 °C, and the detector temperature was set to 250 °C. About 5.0 μL of sample was injected into the GC instrument at a split ratio of 1:20 with nitrogen as the carrier gas at a flow rate of 1.5 mL/min. The content of each SCFA was calculated according to the internal standard method.

### 2.10. Extraction and Analysis of 16s rRNA Gene Sequences

Bacterial genetic DNA was extracted by using a QIA amp DNA Stool Mini kit using primers 338F (5′-ACTCCTACGGGAGGCAGCAG-3′) and 806R (5′-GGACTACHVGGGTWTCTAAT-3′) to amplify the V3-V4 region of the 16s rRNA gene. Sequencing libraries were generated using a TruSeq-DNA PCR-Free sample preparation kit, and pyrosequencing was performed by using the MiSeq PE250 platform. Paired-end read segments were assembled and quality-controlled using FLASH (v 1.2.7) and Qiime (v 1.9.1), respectively. The Uparse (v7.0.1001) algorithm assigned sequences with >97% similarity to the same operational taxonomic units (amplicon sequence variants (ASVs)). Representative sequences of ASVs were also selected based on the principles of its algorithm, and the sequences with the highest frequency of occurrence in the ASVs were selected as representatives of the ASVs. The Silva database (http://www.arb.silva.de/, accessed on 15 January 2023) was used to annotate biological taxonomic information based on the mothur algorithm. α-diversity and β-diversity analyses were performed using Qiime (v1.9.1) and presented using R software (v 2.15.3).

### 2.11. Statistics and Analysis of Data

Origin 2023 software was used for mapping, and all experiments were carried out in three biological replicates. Data were expressed as the mean ± standard deviation, and one-way ANOVA and Tukey’s post hoc test of variance were performed using SPSS 22.0 statistical software, with *p* < 0.05 indicating a significant difference.

## 3. Results

### 3.1. Simulated Gastric and Small Intestinal Digestion

#### 3.1.1. Production of Reducing Sugar and Total Sugar

Figure 1 shows the steamed buns (Figure 1A) and the BGR equipment (Figure 1B). The lightest color of the steamed bun was produced by RF and the darkest color of the steamed bun was produced by WF. Figure 2 shows the changes in reducing and total sugars after digestion in the stomach and small intestine. In the simulated gastric and small intestinal fluids, the levels of reducing and total sugars increased as digestion progressed. Significant differences were noted in the sugar content of the three types of flour (*p* < 0.05), with RF having the highest sugar content, including 17.7 ± 0.398 and 69.1 ± 0.752 g/L reducing sugar and total sugar, respectively, after digestion in gastric fluid (Figure 2A,C) and 6.7 ± 0.104 and 14.9 ± 0.173 g/L, respectively, after digestion in small intestine fluid (Figure 2B,D). Conversely, WF had the least amount of reducing sugar and total sugars, where the reducing and total sugar contents after digestion by gastric fluid were 15.4 ± 0.131 and 46.4 ± 1.244 g/L, respectively. Meanwhile, the reducing and total sugar contents after digestion by small intestinal fluid were 5.8 ± 0.079 and 13.2 ± 0.173 g/L, respectively.

#### 3.1.2. Ability to Scavenge ABTS Free Radicals

All three types of flour digests showed some scavenging of ABTS radicals after simulated gastric and small intestinal digestion. Among the simulated gastric digests, WF showed the best scavenging effect, followed by MF, whereas RF showed the lowest effect (Figure 3A). At 90 min, the scavenging rate was 20.0% for RF, 24.0% for MF, and 26.7% for WF, which was 1.3 times higher than the scavenging rate of RF and 1.1 times higher than the scavenging rate of MF.

In the simulated small intestine digest, the scavenging rate of ABTS radicals generally showed an increasing trend from 0 h to 2 h and then stabilized after 2 h (Figure 3B). At 2 h, the scavenging of ABTS radicals by RF and MF digest was 26.7% and 28.0%, respectively, and the scavenging of ABTS radicals by WF digest was 29.3%, which was 1.1 times that of RF and 1.0 times that of MF.

### 3.2. In Vitro Fecal Fermentation with a Carbon Source Addition of 0.5%

#### 3.2.1. Carbon Source Consumption

After 12 h of fermentation, the content of reducing sugars and total sugars decreased rapidly, and the consumption rate slowed down in the later stages of fermentation (Figure 4A,B). During the fermentation process, OD_600_ gradually increased (Figure 4C), and the pH first decreased, increased, and, finally, stabilized (Figure 4D). After 48 h of fermentation, the reducing sugar contents of the three types of flour were significantly different (*p* < 0.05), with WF having a reducing sugar content of 0.04 ± 0.010 g/L, which was a decrease of 63.4% and 46.1% compared with RF and MF, respectively. The total sugar content of RF was significantly different from that of MF and WF, with WF having a total sugar content of 4.3 ± 0.168 g/L, which was a decrease of 21.3% and 1.8% compared with RF and MF, respectively. However, OD_600_ and pH were not significantly different among RF, MF, and WF, with OD_600_ values of 1.6 ± 0.047, 1.4 ± 0.051, and 1.6 ± 0.127 and pH of 6.1 ± 0.177, 6.0 ± 0.071, and 6.0 ± 0.071, respectively.

#### 3.2.2. Production of SCFAs

The concentration of acetic and propionic acid increased and then decreased. The acetic acid concentrations of RF and WF were significantly different (*p* < 0.05), with the highest acetic acid concentrations of 19.4 ± 6.119, 16.6 ± 1.367, and 12.1 ± 2.715 mmol/L for RF, MF, and WF, respectively (Figure 5A). Reductions of 37.6% and 27.1% in acetic acid concentrations were observed for WF compared with RF and MF, respectively. The highest concentrations of propionic acid for the three types of flour also differed significantly (Figure 5B), with 8.0 ± 0.104, 9.4 ± 0.256, and 11.5 ± 0.686 mmol/L for RF, MF, and WF, respectively. This phenomenon increased the propionic acid concentration of WF by 42.9% and 21.8% compared with RF and MF, respectively. The concentration of isobutyric acid showed a decreasing trend from 0 h to 24 h and started to increase after 24 h (Figure 5C). At 48 h, the concentrations of isobutyric acid were 0.4 ± 0.017, 0.3 ± 0.031, and 0.3 ± 0.034 mmol/L for RF, MF, and WF, respectively. Compared with RF and MF, the concentration of isobutyric acid in WF changed by −13.9% and 14.4% compared with RF and MF, respectively.

The concentration of butyric acid increased over 48 h, with a slow increase during the first 24 h and a significantly rapid increase thereafter (Figure 5D). After 48 h of fermentation, the butyric acid concentrations of the RF and MF fermentations were significantly different from those of WF, with butyric acid concentrations of 11.8 ± 0.395, 11.1 ± 0.335, and 14.9 ± 0.472 mmol/L for RF, MF, and WF, respectively. The butyric acid concentration of WF increased by 26.4% and 34.0% compared with RF and MF, respectively. The concentrations of isovaleric acid of all three types of flour were low and not significantly different (Figure 5E) at 1.3 ± 0.083, 1.4 ± 0.084, and 1.4 ± 0.080 mmol/L for RF, MF, and WF, respectively. In contrast, the content of valeric acid increased over the fermentation time (Figure 5F). A significant difference in the concentration of valeric acid was found between RF and WF. After 48 h of fermentation, the lowest valeric acid concentration was found in RF, followed by MF, and the highest valeric acid concentration was found in WF. The valeric acid concentrations were 4.0 ± 0.825, 4.6 ± 0.691, and 5.7 ± 0.299 mmol/L for RF, MF, and WF, respectively. Compared with RF and MF, the valeric acid concentrations in WF increased by 42.2% and 24.4%, respectively.

WF had the highest total SCFA concentration and was significantly different from RF and MF (Figure 5G), with SCFA concentrations of 24.8 ± 2.944, 25.8 ± 1.833, and 31.1 ± 0.892 mmol/L for RF, MF, and WF, respectively. Compared with RF and MF, total SCFA in WF increased by 25.2% and 20.7%, respectively. In contrast, no significant difference was found in the branched-chain fatty acids (BCFAs) produced by the three different types of flour that are harmful to human health (Figure 5H).

#### 3.2.3. Changes in Intestinal Microbiota

Intestinal microbiota after 48 h of fermentation were selected for 16s rRNA sequencing analysis [32,33]. The community richness and diversity were expressed by using Shannon and Simpson indices, respectively (Figure 6A,B). The Shannon and Simpson indices of the intestinal microbiota of WF were the largest, followed by MF, and, finally, RF. Principal coordinate analysis (PCoA) and non-metric multidimensional scaling (NMDS) analyses were performed to determine the effect of different types of flour on the overall structure and β-diversity of intestinal microbiota (Figure 6C,D). They could reflect the similarity between samples according to the distance of samples projected on the coordinate axis.

The histogram of relative abundance of species at the phylum level (Figure 7A) showed detailed information on the composition of each group of intestinal microbiota. The dominant phyla comprised Proteobacteria, Firmicutes, Bacteroidota, and Actinobacteriota. The relative abundance of Proteobacteria accounted for more than 53.8% of the total bacterial sequences in all experimental groups and ranged from a high of 58.4% relative abundance in RF to a low of 53.8% in WF. Firmicutes had the highest relative abundance of 35.5% in WF and the lowest of 34.0% in RF. Bacteroidota had the highest relative abundance of 8.2% in WF. WF had the lowest ratio of Firmicutes to Bacteroidota (4.3), whereas RF (5.7) and MF (5.9) had 31.6% and 36.6% higher ratios of Firmicutes to Bacteroidota than WF, respectively. WF could promote the growth of Bacteroidota to some extent based on the change in their relative abundance.

As shown in the clustering heat map (Figure 7B), compared with RF and MF, WF significantly promoted the growth of beneficial microbiota such as *Megamonas* (8.1%), *Subdoligranulum* (3.8%), *Parabacteroides* (2.7%), *Blautia* (1.2%), *Muribaculaceae* (0.9%), and *Phascolarctobacterium* (1.1%). In addition, WF may inhibit the growth of potentially pathogenic microbiota, such as *Enterococcus*, *Escherichia–Shigella*, and *Streptococcus*.

### 3.3. WF In Vitro Fecal Fermentation at Different Additions

#### 3.3.1. Consumption of Carbon Sources

As shown in Section 3.2, the potential beneficial effects of WF on intestinal microbiota were most prominent when the carbon source was added at 0.5%. Therefore, the effects of different additions (0.5%, 1.0%, and 1.5%) of WF on intestinal microbiota were explored.

Significant differences were noted in reducing sugars, total sugars, OD_600_, and pH of the WF at different additions (*p* < 0.05), which are consistent with the results in Section 3.2.1. The highest remaining sugars at all stages of fermentation were found in 1.5% WF (Figure 8A,B), with reducing sugar and total sugar contents of 0.3 ± 0.014 and 19.0 ± 0.277 g/L, respectively. These values were 57.6% and 25.0% higher compared with those of 1.0% WF and 551.2% and 340.9% higher compared with those of 0.5% WF, respectively. The OD_600_ value of the intestinal microbiota increased with increasing WF addition (Figure 8C), with 1.5% WF having an OD_600_ value of 1.9 ± 0.047, which was 23.5% and 11.5% higher than 0.5% and 1.0% WF, respectively. The pH decreased with increasing WF addition (Figure 8D), with 1.5% WF having a pH of 4.4 ± 0.029, which was 27.6% and 3.1% lower than 0.5% and 1.0% WF, respectively.

#### 3.3.2. Production of SCFAs

Of the SCFAs measured, acetic acid (Figure 9A), propionic acid (Figure 9B), and butyric acid (Figure 9D) had the highest concentrations, whereas isovaleric acid (Figure 9E), valeric acid (Figure 9F), and isobutyric acid (Figure 9C) had the lowest concentrations. With increasing WF addition, the concentration of acetic acid kept increasing, whereas that of propionic acid, butyric acid, and isovaleric acid kept decreasing. Significant differences were observed in the concentrations of acetic acid and isovaleric acid among the three WF additions (*p* < 0.05). The WF acetic acid concentration at 1.5% was 29.5 ± 3.554 mmol/L, which was an increase of 1167.5% and 20.2% over 0.5% and 1.0% WF, respectively. The isovaleric acid concentration of 1.5% WF was 0.2 ± 0.053 mmol/L, which was 84.7% and 50.2% lower compared with that of 0.5% and 1.0% WF, respectively. Although no significant differences were found in the concentration of SCFAs after fermentation at different WF additions (Figure 9G), 1.5% WF had the highest SCFA concentration of 34.6 ± 4.501 mmol/L and the lowest BCFA concentration (Figure 9H) at 0.4 ± 0.059 mmol/L, which was 79.6% and 29.8% lower than 0.5% and 1.0% WF, respectively.

#### 3.3.3. Changes in Intestinal Microbiota

The α-diversity among different WF additions was not significant, and only the β-diversity showed significant results. As shown in the PCoA and NMDS plots (Figure 10A,B), the distance between different additions of WF samples and control samples was far. The three samples at 1.5% WF addition were more tightly clustered compared with those at 0.5% and 1.0% WF. At the phylum level, the dominant groups in each group were Firmicutes, Proteobacteria, Bacteroidota, and Actinobacteriota (Figure 10C). Proteobacteria accounted for over 38.8% of the microbiome. The relative abundance of Bacteroidota at different WF additions was 8.2%, 5.4%, and 11.7%, indicating that 1.5% WF significantly promoted the growth of Bacteroidota. At the genus level, 1.5% WF significantly inhibited the growth and reproduction of potentially pathogenic microbiota, such as *Streptococcus*, *Enterococcus*, *Klebsiella*, and *Prevotella*, compared with 0.5% and 1.0% WF additions (Figure 9D). Moreover, 1.5% WF had the highest clusters of the most beneficial microbiota, including *Bifidobacterium* (1.8%), *Parasutterella* (2.5%), *Blautia* (7.2%), *Lactobacillus* (4.5%), *Parabacteroides* (2.7%), *Subdoligranulum* (7.5%), *Alistipes* (2.6%), and *Bacteroides* (4.5%), among others.

To further identify bacterial taxa with differences in abundance between different WF additions, linear discriminant analysis effect size analysis was performed (Figure 10E) to screen for significantly enriched bacterial groups at the genus level and identify seven genera with significant differences in abundance. Among the different samples, the abundance of *Enterococcus* was higher at the 0.5% WF addition; *Streptococcus* and *Sutterella* were higher at the 1.0% WF addition; and *Lactobacillus*, *Parasutterella*, *Alistipes*, and *Bifidobacterium* were higher at the 1.5% WF addition.

## 4. Discussion

During simulated in vitro digestion, the steamed buns were fully degraded by digestive enzymes, and significant differences were found in the sugar contents of RF, MF, and WF (*p* < 0.05). WF had the lowest sugar contents and the highest scavenging capacity for ABTS free radicals. An important amount of phenolic compounds, mainly ferulic acid, was linked to the dietary fiber, and this may explain why WF had marked antioxidant activity [34]. RF is finely processed to remove the bran and germ structures, retaining only the internal structure of the endosperm, which is high in sugar, whereas WF is not finely processed and retains the bran, which contains high levels of dietary fiber and other active substances. To date, the recognized beneficial active components in whole wheat mainly include dietary fiber, flavonoids, phenolic acids, alkylresorcinols, benzoxazinoids, phytosteroids, carotenoids, and other minor components [35]. Dietary fiber is a non-digestible carbohydrate that cannot be digested and broken down by digestive enzymes in the stomach and small intestine, so WF has lower-sugar contents than RF and MF [10]; the active substances have an antioxidant effect, so the WF scavenging rate of free radicals is the highest [34,35,36].

In in vitro fecal fermentation with a carbon source addition of 0.5%, the intestinal microbiota made full use of the nutrients, grew and multiplied well, and produced acid metabolism vigorously. SCFAs have been widely studied in human health [10,37,38]. An increase in total SCFA is identified as one of the objectives of dietary interventions. Among them, acetic acid helps maintain a stable intestinal environment and nourishes other beneficial genera of microbiota in the intestine [39]. Propionic acid inhibits cholesterol synthesis, reduces fat storage, and possesses anti-cancer and anti-inflammatory properties [40]. Butyric acid is a metabolite of intestinal microbiota and plays an important role in maintaining the integrity of the colonic epithelium. It even has the potential to improve brain health [41]. Although no significant differences were observed in OD_600_, pH, and BCFA concentrations among RF, MF, and WF, WF resulted in the highest cumulative production of butyric acid and the highest cumulative production of SCFAs, and it was significantly different from RF and MF. These results indicate that WF contained more dietary fiber than RF and MF. Dietary fiber can provide an anaerobic fermentation substrate for the growth and metabolism of intestinal microbiota, resulting in an increase in SCFA production [10].

There is growing evidence from human and animal studies that suggest whole cereal consumption can affect intestinal microbiota composition [36]. The majority of antioxidants, such as phenolic acids, in whole cereals, are bound to dietary fiber components, precluding their absorption in the stomach and small intestine [18]. Therefore, WF had the strongest ability to clear ABTS. However, microbial-derived esterases in the large intestine could release a portion of these compounds from their dietary fiber conjugates, whereupon they are rapidly metabolized, thereby regulating the structure of intestinal microbiota [12]. The relevant regulatory mechanisms need to be further studied. The intestinal microbiota after WF fermentation showed greater α-diversity than RF and MF. Thus, consumption of WF could maintain the intestinal community diversity of intestinal microbiota. Using weighted unifrac-distance-based PCoA, the results show that the different three types of flour promoted changes in the structure of the intestinal microbiota. The control samples were farther away from the samples of the three flour samples, indicating that grouping was meaningful. After adding three different kinds of flour for fermentation, the species composition changed significantly compared with the control. The RF and WF samples were the most distant from each other, indicating a significant change in species composition and community structure similarity after fermentation.

Experimental studies on humans revealed that weight loss is associated with low ratios of Firmicutes and Bacteroidota and high ratios of Firmicutes and Bacteroidota in obese individuals [17,42,43]. In addition, this ratio may be correlated with the metabolic potential of intestinal microorganisms. Obesity control may be facilitated by either increasing the abundance of Bacteroidota or decreasing the abundance of Firmicutes. Although no significant differences in the regulation of major phyla were found among the three types of flour after fermentation, WF had the lowest ratio of Firmicutes to Bacteroidota; this finding is consistent with the results of the research of Parnel et al. on prebiotic fiber, suggesting that whole cereals have the potential to modulate the structure of intestinal microbiota and improve obesity [44].

WF increased the relative abundance of beneficial microbiota, such as *Megamonas*, *Subdoligranulum*, *Blautia*, and *Phascolarctobacterium*. Among them, *Megamonas* can ferment various carbohydrates to produce beneficial organic acids, such as acetic acid, propionic acid, and lactic acid, which are associated with symptoms of frailty in the elderly [45]. *Subdoligranulum* can ferment carbohydrates to produce butyric acid, which strengthens the intestinal barrier and the body’s immune system, potentially helping reduce the risk of colon cancer [46]. *Blautia* may help alleviate inflammatory and metabolic diseases and have antimicrobial activity against specific microorganisms [47]. *Phascolarctobacterium* may be a key regulator of the dynamic balance of the intestinal microbiota, potentially helping predict the risk of obesity and preventing *Clostridium* difficile colonization [48]. In conclusion, the consumption of dietary-fiber-rich WF could produce more SCFAs, effectively inhibit the growth of harmful microbiota, and promote the growth of beneficial microbiota in human intestinal microbiota compared with RF and MF.

Studies have suggested that dietary composition and intake may affect the composition of the intestinal microbiota and the concentration of SCFAs [49,50]. At three different additions of WF (0.5%, 1.0%, and 1.5%), significant differences in OD_600_ and pH were found among them. No significant differences were found in the cumulative production of SCFAs among the three, but WF at 1.5% resulted in the highest cumulative production of SCFAs and the lowest cumulative production of BCFAs.

A double-blind clinical trial on humans reported that intake of whole wheat or a diet with added bran can adjust intestinal microbiota and increase the number of *Bifidobacterium* [51]. A human experiment showed that whole-grain wheat intake increased the number of *Bifidobacterium* and *Lactobacillus* in feces [52]. In this paper, 1.5% WF had the best reproducibility, the most similar species composition structure, and the highest relative abundance of beneficial microbiota, such as *Bifidobacterium*, *Parasutterella*, *Lactobacillus*, *Alistipes*, and *Bacteroides*. Among them, *Bifidobacterium* can regulate intestinal microbiota and is an important beneficial intestinal microorganism with various physiological effects, such as the synthesis of vitamins, stimulation of immune function, improvements in host resistance to infection, and anti-tumor functions. *Parasutterella* is a potential probiotic that may be involved in maintaining bile acid homeostasis and cholesterol metabolism [53]. *Lactobacillus* is effective in maintaining the regeneration of the intestinal epithelium and homeostasis in vivo, as well as repairing intestinal mucosal damage after pathological injury [54]. *Alistipes* may have protective effects against chronic diseases such as liver fibrosis, cancer immunotherapy, and cardiovascular disease [55]. Therefore, the intake of 1.5% WF had a better effect on intestinal microbiota and produced a more beneficial effect on human health than the other tested percentages.

## 5. Conclusions

In conclusion, the consumption of WF increased the production of SCFAs in the gut, modulated the structure of the intestinal microbiota, promoted the proliferation of beneficial genera, and inhibited the proliferation of potentially pathogenic bacteria. The association between many chronic diseases and intestinal microbiota makes it necessary to evaluate the effects of whole and refined cereals on intestinal microbiota. These results support the rational dietary theory that humans should increase their consumption of whole cereals and offer insights into the development of new functional foods derived from wheat. However, this study has only been explored in an in vitro model and needs to be further validated and explored through sensorial analysis, phenolic metabolites analysis, and animal experiments.

## Figures and Tables

**Figure 1 foods-12-02847-f001:**
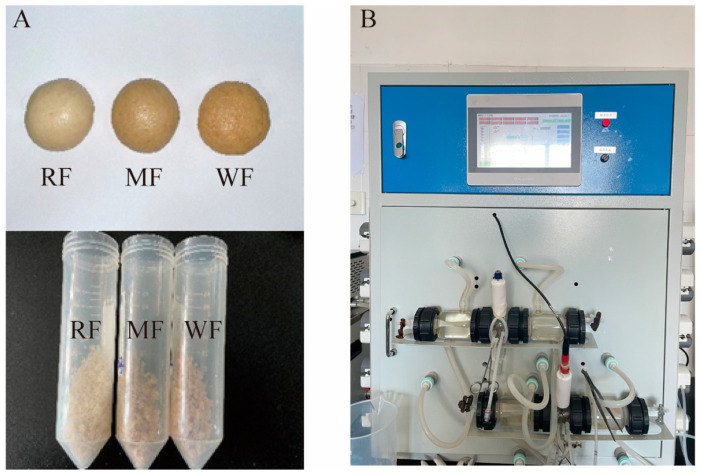
(**A**) Steamed buns made from different types of flour; (**B**) BGR equipment.

**Figure 2 foods-12-02847-f002:**
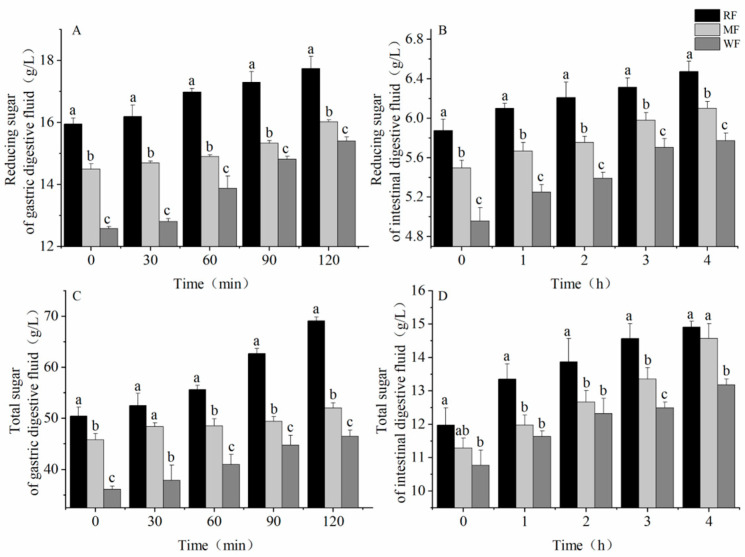
(**A**) Reducing sugar content after gastric digestion; (**B**) total sugar content after gastric digestion; (**C**) reducing sugar content after small intestine digestion; (**D**) total sugar content after small intestine digestion. Values are means ± SEM. Different letters indicate a significant (*p* < 0.05) difference using repeated-measures ANOVA and Tukey’s post hoc test.

**Figure 3 foods-12-02847-f003:**
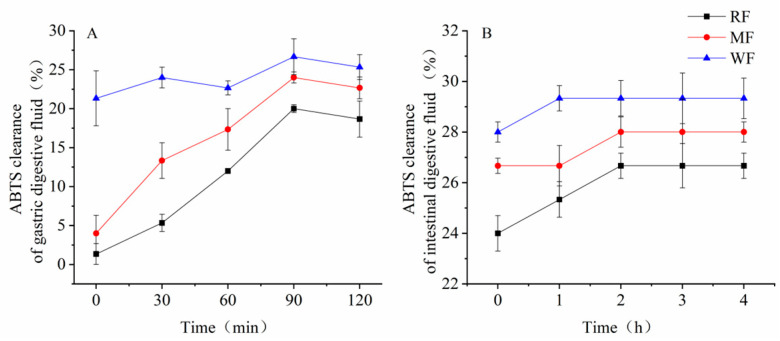
(**A**) ABTS free radical scavenging capacity after gastric digestion; (**B**) ABTS free radical scavenging capacity after small intestine digestion. Values are means ± SEM. Different letters indicate a significant (*p <* 0.05) difference using repeated-measures ANOVA and Tukey’s post hoc test.

**Figure 4 foods-12-02847-f004:**
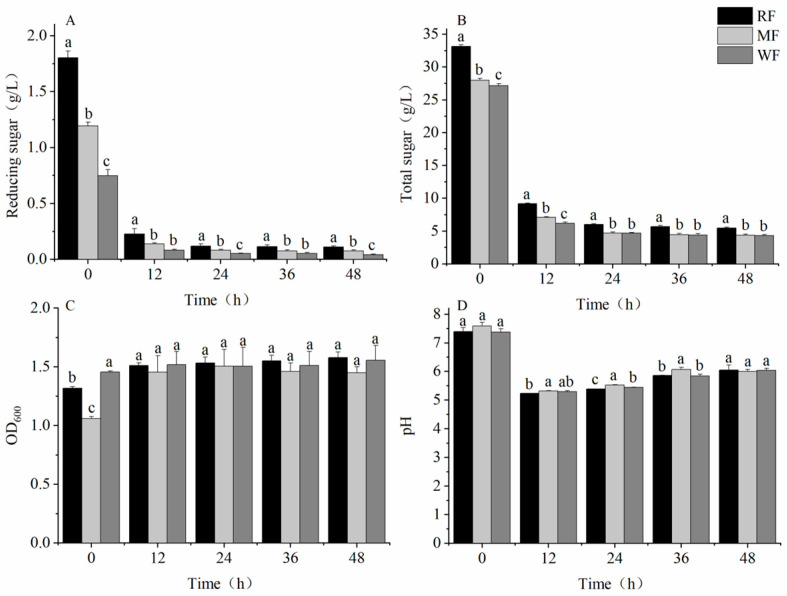
Changes in parameters of different types of flour after fecal fermentation in vitro. (**A**) Reducing sugar content; (**B**) total sugar content; (**C**) OD_600_; (**D**) pH. Values are means ± SEM. Different letters indicate a significant (*p* < 0.05) difference using repeated-measures ANOVA and Tukey’s post hoc test.

**Figure 5 foods-12-02847-f005:**
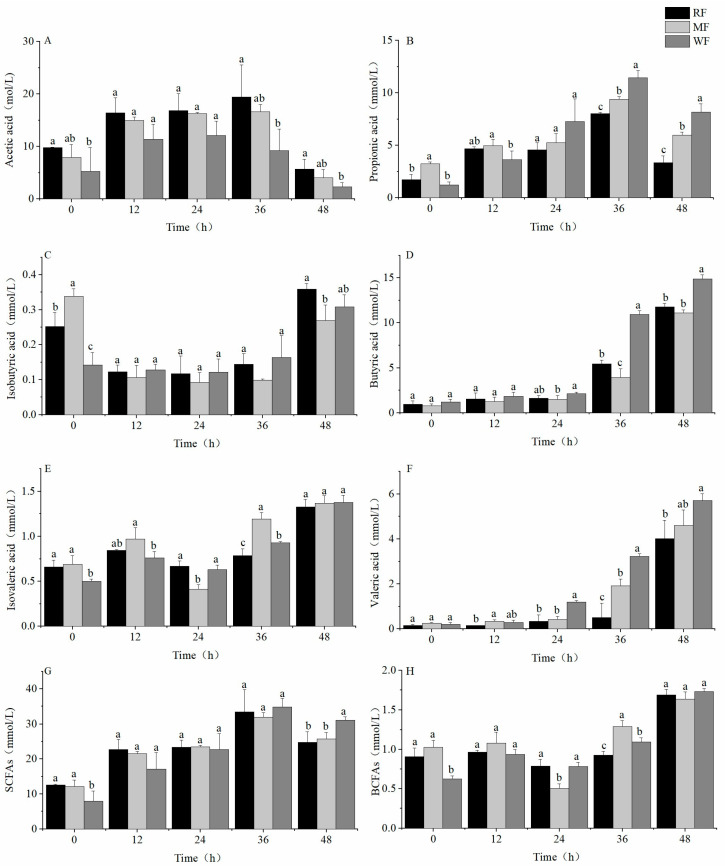
Production of SCFAs with a carbon source addition of 0.5%. (**A**) Acetic acid; (**B**) propionic acid; (**C**) isobutyric acid; (**D**) butyric acid; (**E**) isovaleric acid; (**F**) valeric acid; (**G**) SCFAs; (**H**) BCFAs. Values are means ± SEM. Different letters indicate a significant (*p* < 0.05) difference using repeated-measures ANOVA and Tukey’s post hoc test.

**Figure 6 foods-12-02847-f006:**
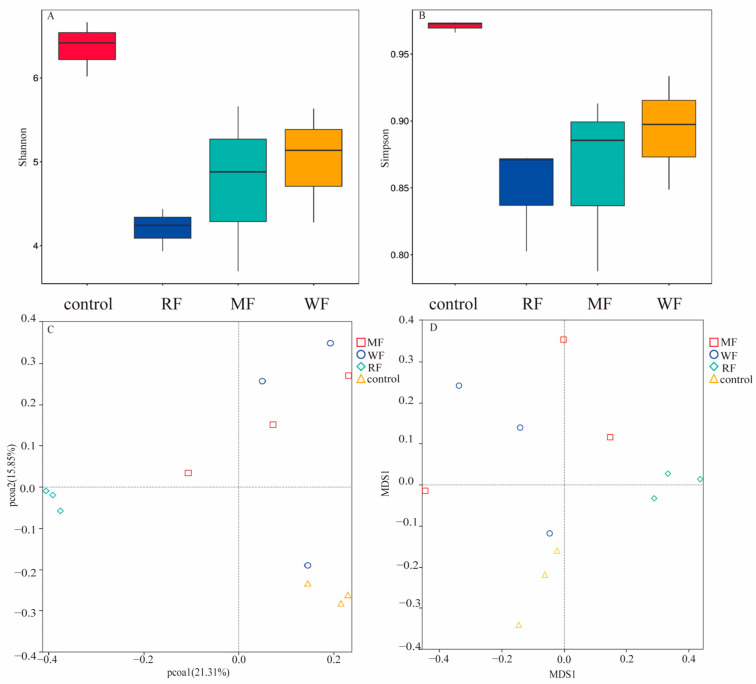
α-diversity and β-diversity of the intestinal microbiota; (**A**) Shannon indices; (**B**) Simpson indices; (**C**) weighted unifrac-distance-based PCoA; (**D**) weighted unifrac-distance-based NMDS. PCoAs 1 and 2, MDS1 and 2 represent the major axes of variation among objects in a 2D space. Each dot in the graph represents a sample, and dots of different colors indicate different groups. The percentages in the coordinate brackets represent the proportions of the sample variance data (the distance matrix) that the corresponding coordinate axis can interpret.

**Figure 7 foods-12-02847-f007:**
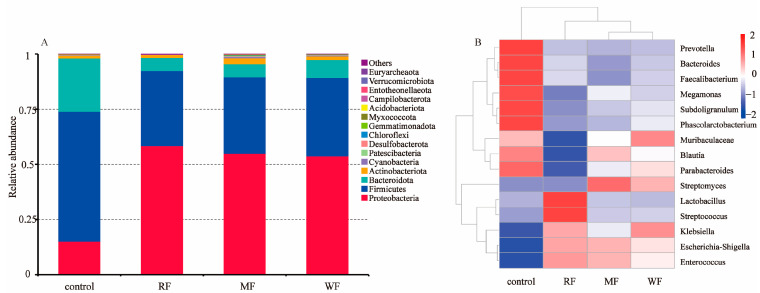
(**A**) Histogram of relative abundance at the phylum level; (**B**) heat map of clustering at the genus level.

**Figure 8 foods-12-02847-f008:**
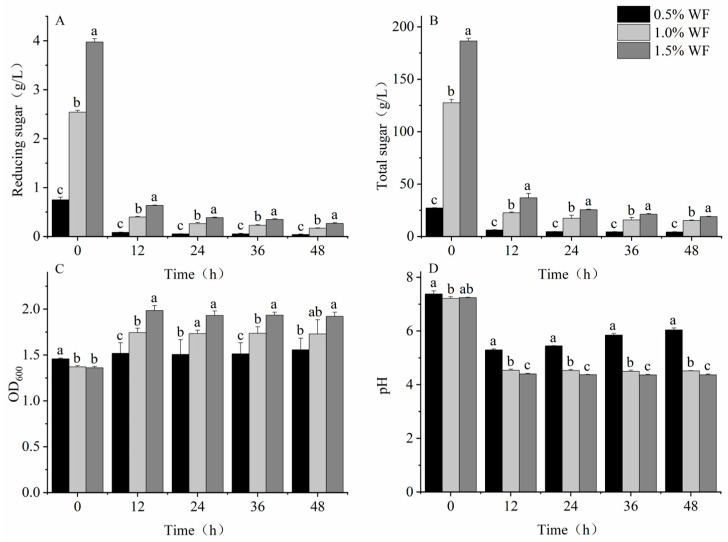
Changes in each parameter after in vitro fecal fermentation of WF at different additions. (**A**) Reducing sugar content; (**B**) total sugar content; (**C**) OD_600_; (**D**) pH. Values are means ± SEM. Different letters indicate a significant (*p* < 0.05) difference using repeated-measures ANOVA and Tukey’s post hoc test.

**Figure 9 foods-12-02847-f009:**
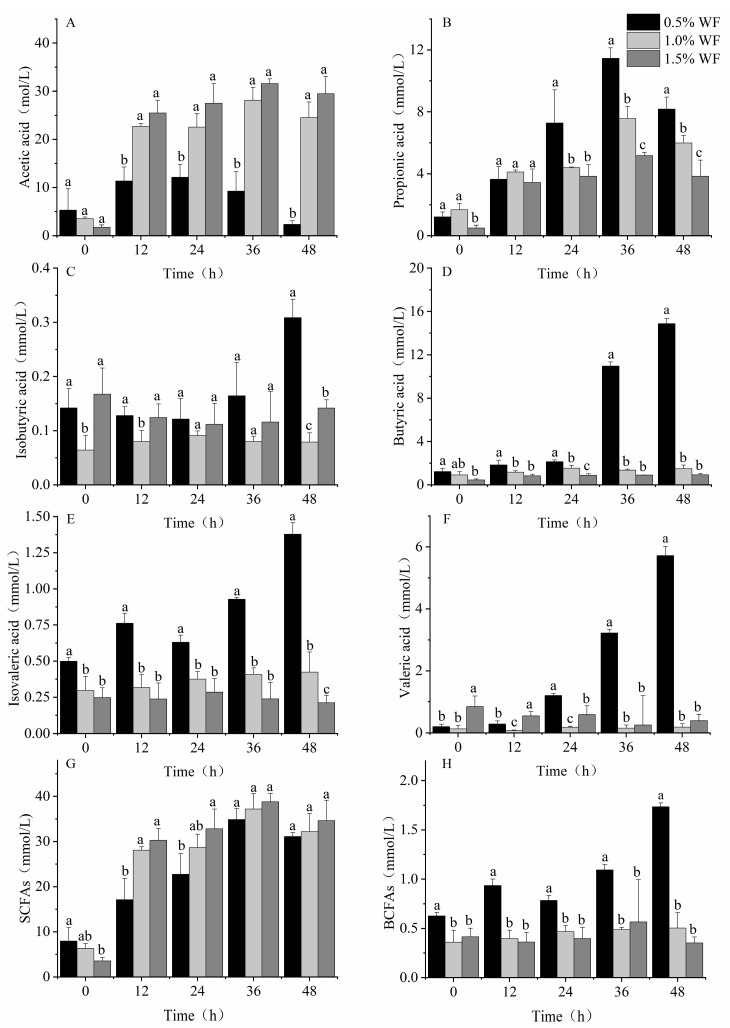
Production of SCFAs at different additions of WF. (**A**) Acetic acid; (**B**) propionic acid; (**C**) isobutyric acid; (**D**) butyric acid; (**E**) isovaleric acid; (**F**) valeric acid; (**G**) SCFAs; (**H**) BCFAs. Values are means ± SEM. Different letters indicate a significant (*p* < 0.05) difference using repeated-measures ANOVA and Tukey’s post hoc test.

**Figure 10 foods-12-02847-f010:**
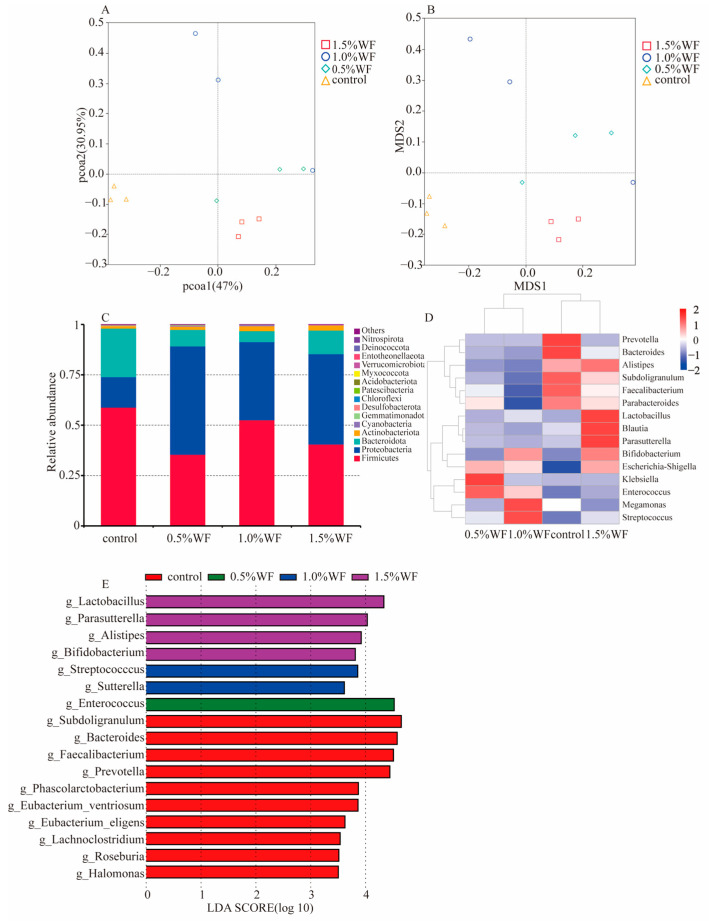
(**A**) Weighted unifrac-distance-based PCoA; (**B**) weighted unifrac-distance-based NMDS; (**C**) histogram of relative abundance at the phylum level; (**D**) heat map of clustering at the genus level; (**E**) Histogram of LDA SCORE distribution.

## Data Availability

All raw data for 16s rRNA sequences were deposited into the NCBI Sequence Read Archive (accession number: PRJNA982619).

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
