# Peer review of "A Comparative Study of the Effects of Whole Cereals and Refined Cereals on Intestinal Microbiota"

_foods, 2023, doi:10.3390/foods12152847_

Round 1

Reviewer 1 Report

In the Introduction section, studies about the effect on intestinal microbiota of whole wheat introduction in other food products (noodles, bread, etc.) should be mentioned, including in vitro, animal and clinical studies.

In line 103, the amount of bun used in vitro digestion simulation should be indicated.

In line 106, regarding mouth digestion, it is not indicated if mastication of the bun was simulated. If yes, indicate which procedure was used; if not, state how the simulation of mastication is not essential in this kind of product.

Which faecal fermentation time was used to Analyse the changes in intestinal microbiota and why this time was selected?

A analysis of a control through the time regarding pH and SCFAs were important to evaluate the inoculum.

In Figure 5, the graphic E, G, e H  miss some letters of statistical analysis. Verify. Besides that, explain why the hours of the samples are different from the other analyses regarding in vitro colon fermentation (Reducing sugar content; Total sugar content; OD600 pH.).

In line 438, explain what you mean by active substances. Besides that, bibliographic references should be used in line 441 to support the affirmation stated.

In line 457, bibliographic references should be cited to support the conclusion established.

In the discussion section, the effect on intestinal microbiota obtained in other studies where whole wheat flour replaced refined flour should be cited and used to improve the quality of the discussion, including in vitro, animal and clinical studies.

In the discussion section, the importance of phenolic compounds in whole wheat flour to improve the prebiotic effect of this flour should be described.

In conclusion, a clear indication of which further studies were necessary should be indicated, namely sensorial analysis, phenolic metabolites analysis and other studies which authors think to be relevant. 

Minor editing of English language required

Reviewer 2 Report

Cereals are one of the most important foods that humans rely on to sustain basic life activities, and different types of cereals have different effects on human health and this study investigated the effects of different steamed buns on the intestinal microbiota. Basically this research is in an interesting topic.

1. "WF and RF were purchased at a local supermarket." need details, what are the contents of WF and RF? why is it called as cereal? what is the content? must be specified in detail.

2. "Statistics and analysis of data" for intestinal microbiota data has not been mentioned which software to use? with what analysis? must be detailed.

3. Why does "Measurement of SCFAs" only use GC? is it not combined with MS (GC-MS)?

4. Heatmap of Pearson correlation between the gut microbiome and the ABTS activity should be performed and presented in the manuscript.

5. At the end of the discussion section, a discussion regarding the limitations of this study should be given; see this study is only In Vitro which needs further study to see the efficacy at an advanced level.

There are no big problems in English, but grammar needs to be rechecked.
